# Neural Network Regression with Beta, Dirichlet, and Dirichlet-Multinomial Outputs

## Abstract

We propose a method for quantifying uncertainty in neural network regression models when the targets are real values on a $d$-dimensional simplex, such as probabilities. We show that each target can be modeled as a sample from a Dirichlet distribution, where the parameters of the Dirichlet are provided by the output of a neural network, and that the combined model can be trained using the gradient of the conditional likelihood. This approach provides interpretable predictions in the form of multidimensional distributions, rather than point estimates, from which one can obtain confidence intervals or quantify risk in decision making. Furthermore, we show that the same approach can be used to model targets in the form of empirical counts as samples from the Dirichlet-multinomial compound distribution. In experiments, we verify that our approach provides these benefits without harming the performance of the point estimate predictions on two diverse applications: (1) distilling deep convolutional networks trained on CIFAR-100, and (2) predicting the location of particle collisions in the XENON1T Dark Matter detector.

## 1 Introduction

Artificial neural networks are typically trained by maximizing the conditional likelihood of output targets given input features. Each target is modeled as a sample from a distribution $p(y|x)$ parameterized by the output activity of the neural network, where the choice of parametric distribution is implied by the choice of objective function. Thus, the support of the probability distribution should match the target space, but in practice, this is often not the case.

Today, the vast majority of neural network output layers implicitly model the targets as samples from one of four distributions: a binomial, a categorical, a Gaussian, or a Laplacian distribution — respectively corresponding to the binomial cross-entropy loss, multi-class cross-entropy loss, mean squared error, and mean absolute error. These distributions are commonly used even when the target space does not match the support, because the gradient calculations for these distributions are simple (and easy to compute) when paired with the appropriate output layer activation functions. These distributions dominate to such a degree that few alternatives are even available in most common deep learning software packages such as *Keras* (Chollet et al., 2015) and *PyTorch* (Paszke et al., 2017).

Alternatives do exist — using neural networks to parameterize more complex distributions is not new. The standard regression approach can be generalized to a *heteroskedastic* Gaussian output layer (Nix & Weigend, 1994; Williams, 1996), where the neural network predicts both a mean and a variance for each target. Multi-model distributions can be modeled with a mixture density (Bishop, 1994). And more recently, the Gamma output layer was proposed to model targets in $\mathbb{R}_{>0}$ (Ng et al., 2017). In principle, any parametric distribution with well-defined gradients could serve as a probabilistic prediction at the output of a neural network model.

The approach proposed here is simpler than the one taken by Conditional Variational Autoencoders (CVAEs) (Kingma & Welling, 2013; Sohn et al., 2015). While CVAEs can, in theory, model arbitrary high-dimensional conditional distributions, computing the exact conditional likelihood of a target requires marginalizing over intermediate representations, making exact gradient calculations

intractable. Thus, training a CVAE requires approximating the gradients through sampling. In this work we show that restricting the output to a particular class of distributions, namely the Dirichlet or Dirichlet-multinomial compound distributions, enables a calculation of the exact likelihood of the targets and the exact gradients.

Interpreting the output of a neural network classifier as a probability distribution has obvious benefits. One can derive different point estimates, define confidence intervals, or integrate over possible outcomes — a necessity for managing risk in decision making. Potentially, it could also lead to better learning — matching the output support to the target space essentially constrains the learning problem by incorporating outside knowledge. Allowing the network to output "uninformative" distributions — e.g. a uniform distribution over the support — could make training faster by allowing the network to focus on the easiest training examples first — a self-guided form of curriculum learning.

In the present work, we derive gradients for the Beta distribution, Dirichlet distribution, and Dirichlet-multinomial compound distribution. We then propose activation functions that stabilize numerical optimization with stochastic gradient descent. Finally, we demonstrate through experiments that this approach can be used to model three common types of targets: (1) targets over the multivariate simplex, (2) real-valued scalar targets with lower and upper bounds, and (3) non-negative integer-valued counts (samples from the Dirichlet-multinomial compound distribution). The experiments demonstrate that our approach provides interpretable predictions with *learned* uncertainty, without decreasing the performance of the point estimates.

## 2 DIRICHLET OUTPUT LAYERS

### 2.1 TARGETS ON THE MULTIDIMENSIONAL SIMPLEX

Consider a supervised learning scenario, in which the goal is to model the relationship between a set of input, target pairs $(\boldsymbol{x}^{(t)}, \boldsymbol{y}^{(t)})$, where $\boldsymbol{y}^{(t)}$ lies on the $d$-simplex $\Delta^d$, i.e. $\boldsymbol{y}^{(t)} \in \mathbb{R}^d$ with $y_i^{(t)} > 0 \,\forall\, i \in \{1, \ldots, d\}$ and $\sum_{i=1}^d y_i^{(t)} = 1$. We can construct a neural network that takes in $\boldsymbol{x}^{(t)}$ and outputs a length-$d$ vector $\boldsymbol{\alpha}^{(t)} = <\alpha_1^{(t)}, \ldots, \alpha_d^{(t)}>$, with $\alpha_i^{(t)} \in (0, \infty)$, that parameterizes a Dirichlet distribution over $\Delta^d$ with probability density function

$$p_{\boldsymbol{\alpha}}(\boldsymbol{z}) = \frac{\prod_{i=1}^d z_i^{\alpha_i - 1}}{B(\boldsymbol{\alpha})} \tag{1}$$

$$B(\boldsymbol{\alpha}) = \frac{\prod_{i=1}^d \Gamma(\alpha_i)}{\Gamma(\sum_{i=1}^d \alpha_i)} \tag{2}$$

where $\Gamma$ is the Gamma function, which generalizes the factorial function to real values. Thus, given a set of neural network weights and some input $\boldsymbol{x}^{(t)}$, we have a conditional density function over the domain of the target $\boldsymbol{y}^{(t)}$. The network can be trained to minimize the negative log-likelihood (NLL) of the training set, $-\sum_t \ln p_{\boldsymbol{\alpha}^{(t)}}(\boldsymbol{y}^{(t)})$, using gradient descent. Dropping the example index $t$ for clarity, we write the log-likelihood for a single example as

$$\begin{aligned} \ln(p_{\boldsymbol{\alpha}}(\boldsymbol{y})) &= \ln\left(\prod_{i=1}^d y_i^{\alpha_i - 1}\right) - \ln\left(\frac{\prod_{i=1}^d \Gamma(\alpha_i)}{\Gamma(\sum_{i=1}^d \alpha_i)}\right) \\ &= \sum_{i=1}^d \ln(y_i^{\alpha_i - 1}) + \ln(\Gamma(\sum_{i=1}^d \alpha_i)) - \sum_{i=1}^d \ln(\Gamma(\alpha_i)) \\ &= \sum_{i=1}^d (\alpha_i - 1)\ln(y_i) + \ln(\Gamma(\sum_{i=1}^d \alpha_i)) - \sum_{i=1}^d \ln(\Gamma(\alpha_i)). \end{aligned} \tag{3}$$

The gradient w.r.t. the network output $\alpha_i$ is then

$$\frac{\partial}{\partial \alpha_i} \ln(p_{\boldsymbol{\alpha}}(\boldsymbol{y})) = \ln(y_i) + \frac{\partial}{\partial \alpha_i} \ln(\Gamma(\sum_{j=1}^{d} \alpha_j)) - \frac{\partial}{\partial \alpha_i} \ln(\Gamma(\alpha_i))$$

$$= \ln(y_i) + \frac{\partial(\sum_{j=1}^{d} \alpha_j)}{\partial \alpha_i} \psi(\sum_{j=1}^{d} \alpha_j) - \psi(\alpha_i)$$

$$= \ln(y_i) + \psi(\sum_{j=1}^{d} \alpha_j) - \psi(\alpha_i) \qquad (4)$$

where $\psi$ is the digamma function $\psi(x) = \frac{\partial}{\partial x} \ln(\Gamma(x))$, and the gradient for a particular target is

$$\frac{\partial}{\partial \alpha_i} \ln(p_{\boldsymbol{\alpha}}(\boldsymbol{z} = \boldsymbol{y})) = \ln(y_i) + \psi(\sum_{j=1}^{d} \alpha_j) - \psi(\alpha_i) \qquad (5)$$

Accurate numerical approximations of the digamma function are readily available, and obtaining point estimates from the network output is as simple as computing the mean of the Dirichlet distribution, $E[\boldsymbol{z}] = \frac{\boldsymbol{\alpha}}{\sum_{i=1}^{d} \alpha_i}$, or the mode $\frac{\boldsymbol{\alpha} - 1}{\sum_{i=1}^{d} \alpha_i - d}$ when $\alpha_i > 1 \quad \forall i \in \{1, \dots, d\}$.

The Dirichlet distribution can also be parameterized by a vector $\boldsymbol{\mu} \in \Delta^d$, together with a scalar $\gamma > 0$, where

$$p_{\boldsymbol{\mu}, \gamma}(\boldsymbol{z}) = \frac{\prod_{i=1}^{d} z^{\alpha_i - 1}}{B(\boldsymbol{\alpha})} \qquad (6)$$

$$\alpha_i = \mu_i \times \gamma \qquad (7)$$

In this alternative parameterization, $\boldsymbol{\mu}$ predicts the expectation of the Dirichlet, and the summation to one could be enforced using a *softmax* activation. This is conceptually similar to the heteroskedastic Gaussian model where the neural network computes the mean and standard deviation of a Gaussian distribution.

## 2.2 TARGETS SAMPLED FROM A DIRICHLET-MULTINOMIAL COMPOUND DISTRIBUTION

The Dirichlet output layer can also be used to model vectors of non-negative integer targets $\boldsymbol{y} \in \mathbb{N}_0^d$ as samples from the Dirichlet-multinomial compound distribution. This allows us to treat each target as a collection of *related* trials, conditioned on the same input. The distribution is parameterized by the Dirichlet parameters $\boldsymbol{\alpha}$ and the number of multinomial trials $n$. The number of trials $n$ can be fixed for each training example, fixed for the entire data set, or treated as a random variable sampled from a conditional probability distribution parameterized by an additional neural network output. Here we assume that $n = \sum_i y_i$ is given for each training example, so that the probability of target $\boldsymbol{y}$ is given by the following:

$$p_{\boldsymbol{\alpha}, n}(\boldsymbol{y}) = \int_{\Delta^d} \mathrm{Mul}_{\boldsymbol{z}, n}(\boldsymbol{y}) \mathrm{Dir}_{\boldsymbol{\alpha}}(\boldsymbol{z}) \mathrm{d}\boldsymbol{z}$$

$$= \int_{\Delta^d} \left( \frac{1}{B(\boldsymbol{y})} \prod_{i=1}^{d} z_i^{y_i} \right) \left( \frac{1}{B(\boldsymbol{\alpha})} \prod_{i=1}^{d} z_i^{\alpha_i - 1} \right) \mathrm{d}\boldsymbol{z}$$

$$= \frac{1}{B(\boldsymbol{\alpha}) B(\boldsymbol{y})} \int_{\Delta^d} \prod_{i=1}^{d} z_i^{y_i + \alpha_i - 1} \mathrm{d}\boldsymbol{z}$$

$$= \frac{B(\boldsymbol{\alpha} + \boldsymbol{y})}{B(\boldsymbol{\alpha}) B(\boldsymbol{y})} \qquad (8)$$

where we use the definition of the multivariate beta function to integrate over the simplex $\Delta^d$ and marginalize out $\boldsymbol{z}$. This leads to the log-likelihood

$$
\begin{aligned}
\ln(p_{\boldsymbol{\alpha},n}(\boldsymbol{y})) &= \ln B(\boldsymbol{\alpha} + \boldsymbol{y}) - \ln B(\boldsymbol{\alpha}) - \ln B(\boldsymbol{y}) \\
&= \ln\left(\frac{\prod_{i=1}^{d} \Gamma(\alpha_i + y_i)}{\Gamma(\sum_{i=1}^{d} \alpha_i + y_i)}\right) - \ln\left(\frac{\prod_{i=1}^{d} \Gamma(\alpha_i)}{\Gamma(\sum_{i=1}^{d} \alpha_i)}\right) - \ln B(\boldsymbol{y}) \\
&= \sum_{i=1}^{d} \ln(\Gamma(\alpha_i + y_i)) - \ln(\Gamma(\sum_{i=1}^{d}(\alpha_i + y_i))) \\
&\quad + \ln(\Gamma(\sum_{i=1}^{d} \alpha_i)) - \sum_{i=1}^{d} \ln(\Gamma(\alpha_i)) - \ln B(\boldsymbol{y})
\end{aligned}
\tag{9}
$$

and the gradient of the log-likelihood w.r.t. network output $\alpha_i$,

$$
\frac{\partial}{\partial \alpha_i} \ln\left(p_{\boldsymbol{\alpha},n}(\boldsymbol{y})\right) = \psi(\alpha_i + y_i) - \psi(\sum_{j=1}^{d}(\alpha_j + y_j)) + \psi(\sum_{j=1}^{d} \alpha_j) - \psi(\alpha_i).
\tag{10}
$$

## 2.3 Univariate Targets with Lower and Upper Bounds

When the targets are real-valued scalars with lower and upper bounds, we can shift and rescale the values to be in the range $[0, 1]$ and model the target as a sample from a Beta distribution. The Beta distribution is the $d = 2$ case of the Dirichlet, and can be used to predict univariate targets $y \in [0, 1]$. It can be parameterized by a neural network that outputs two values $\alpha, \beta > 0$, with

$$
p_{\alpha,\beta}(z) = z^{\alpha-1}(1 - z)^{\beta-1} \frac{\Gamma(\alpha\beta)}{\Gamma(\alpha)\Gamma(\beta)}
\tag{11}
$$

with log-likelihood

$$
\begin{aligned}
\ln\left(p_{\alpha,\beta}(z)\right) = {}&(\alpha - 1)\ln(z) + (\beta - 1)\ln(1 - z) \\
&+ \ln(\Gamma(\alpha\beta)) - \ln(\Gamma(\alpha)) - \ln(\Gamma(\beta))
\end{aligned}
\tag{12}
$$

and gradients

$$
\frac{\partial}{\partial \alpha} \ln\left(p_{\alpha,\beta}(z)\right) = \ln(z) + \psi(\alpha\beta) - \psi(\alpha)
\tag{13}
$$

$$
\frac{\partial}{\partial \beta} \ln\left(p_{\alpha,\beta}(z)\right) = \ln(1 - z) + \psi(\alpha\beta) - \psi(\beta).
\tag{14}
$$

Alternatively, we can parameterize the Beta using two scalar network outputs: $\mu \in [0, 1]$ and $\gamma > 0$.

$$
p_{\mu,\gamma}(z) = y^{\alpha-1}(1 - z)^{\beta-1} \frac{\Gamma(\alpha\beta)}{\Gamma(\alpha)\Gamma(\beta)}
\tag{15}
$$

$$
\alpha = \mu \times \gamma
\tag{16}
$$

$$
\beta = \gamma - \mu \times \gamma
\tag{17}
$$

In this parameterization, $\mu = \frac{\alpha}{\alpha+\beta} = E[z]$ is the expectation of the distribution, and $\gamma = \alpha + \beta$ controls the width of the density function.

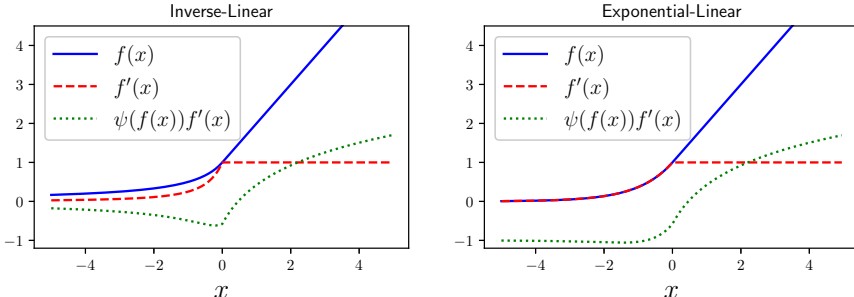

Figure 1: The choice of activation function in the output layer of a neural network affects the shape of the objective function. To stabilize learning in a Dirichlet output layer, we propose that the activation function $f$ should approach 0 asymptotically as $x \to -\infty$, and that $\frac{\partial}{\partial x}\psi(f(x)) = \psi(f(x))f'(x)$ should be bounded. Two such functions are the *Inverse-Linear* (left) and *Exponential-Linear* (right) piecewise functions.

## 3 STABILITY

In order to stabilize learning with stochastic gradient descent, the activation function should be chosen carefully to shape the objective function. For the models proposed here, we need an activation function with a strictly positive range, and that asymptotically approaches zero (or some minimum value $\epsilon > 0$) as $x \to -\infty$. Moreover, the digamma terms in the gradient of the Dirichlet become large for small $\alpha_i$ ($lim_{x\to 0^+}\psi(x) = -\infty$), so to avoid large gradients (which can destabilize learning), we propose two piecewise activation functions for which $\frac{\partial}{\partial x}\psi(f(x))$ is bounded: the *Inverse-Linear* (*IL*) and *Exponential-Linear* (*EL*) functions (Figure 1). The latter is simply a strictly-positive variant of the popular *Exponential Linear Unit*, or *ELU* (Clevert et al., 2015).

$$IL(x) = \begin{cases} \frac{1}{1-x} & \text{for} \quad x < 0, \\ x+1 & \text{for} \quad x \geq 0 \end{cases} \tag{18}$$

$$EL(x) = \begin{cases} e^x & \text{for} \quad x < 0, \\ x+1 & \text{for} \quad x \geq 0 \end{cases} \tag{19}$$

We also propose an activation function that saturates at hyper-parameter value $\tau > 1$, the *Exponential-Tanh* (*ET*):

$$ET_\tau(x) = \begin{cases} e^x & \text{for} \quad x < 0, \\ (\tau-1) \times tanh(\frac{x}{\tau-1}) + 1 & \text{for} \quad x \geq 0 \end{cases} \tag{20}$$

Each of these activation functions were tested in experiments. We found that in both parameterizations of the Dirichlet output, using the *IL* activation resulted in a more stable learning trajectory than with the *EL*, presumably because $\frac{1}{1-x}$ approaches zero more slowly than $e^x$. We also observed overflow errors when some $\alpha_i$ became very small. We found that this could be avoided through weight regularization or by adding a small stability factor $\epsilon$ to the output activation function.

Even with these activation functions, the increased flexibility of the models could lead to unexpected behavior. The model is able to concentrate probability mass at a particular target value, which could allow a network to devote its limited capacity to maximizing the likelihood of a single example. While this was not observed in our experiments, additional regularization and hyper-parameter tuning might be required for some applications.

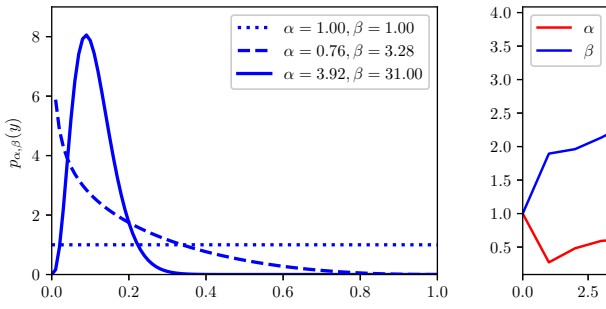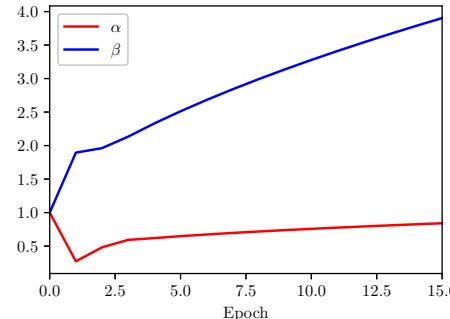

Figure 2: Example evolution of the probability density function for a simple neural network with a Beta output, trained to optimize the log-likelihood of a single data point with $y = 0.1$. The density functions are given by $p_{\theta_1,\theta_2}(y) = y^{\alpha-1}(1-y)^{\beta-1}/B(\alpha,\beta)$ where $\alpha = EL(\theta_1)$ and $\beta = EL(\theta_2)$. Initializing $\theta_1$ and $\theta_2$ to zero gives a uniform distribution (dotted line). After 10 iterations of gradient descent updates to $\theta_1, \theta_2$, the probability mass accumulates in the left corner (dashed line), with $\alpha < 1$. Then $\alpha$ increases and the mode of the distribution becomes non-zero when $\alpha > 0$. At 1000 iterations, the mode is close to the target at $y = 0.1$ (solid line).

# 4 EXPERIMENTS

## 4.1 SIMPLE SIMULATION WITH BETA OUTPUT

As a simple illustration, consider a single data point $y = 0.1$ modeled as a sample from a Beta distribution parameterized by $\alpha = EL(\theta_1)$ and $\beta = EL(\theta_2)$, with the exponential linear activation function $EL$ defined as in Equation 19. We update $\theta_1$ and $\theta_2$ using gradient descent as if they were parameters of a very simple neural network. Figure 2 shows that this model quickly learns to concentrate probability mass at the target.

## 4.2 REGRESSION TASK FOR XENON1T DARK MATTER DETECTOR

For real-valued targets in a bounded interval, one might expect performance to improve when these bounds are incorporated into the model. We tested this idea on a simulated data set from the XENON1T dark matter detection experiment (XENON Collaboration, 2017b), where the task is to predict the $x$ and $y$ locations of individual particle collisions from detector data — essentially a video of the sensor activities over time. The location of each collision is bounded by the dimensions of the detector. The training data consists of 160,000 simulated collision events (XENON Collaboration, 2017a), while another 20,000 events are used for early stopping and validation. The detector has rotational symmetry isomorphic to the cyclic group of order 6, which is accounted for by randomly rotating each example during training. For each event, the neural network input consists of real-valued recordings from 248 detector elements for 1000 time steps, and a neural network predicts both the $x$ and $y$ locations of the event, each normalized to be in the range $[0, 1]$.

A "typical" deep neural network regression model was constructed with an MSE objective and a linear output layer, where one output unit predicted the $x$-location, and another predicted the $y$-location. The architecture and other hyperparameters were optimized using Sherpa (Hertel et al., 2018), including the number and shape of the layers, the learning rate, momentum, and dropout regularization. The best architecture used a two-column siamese network design to process the data from the top and bottom of the detector, with three convolutional layers of shape 300-300-10 followed by two dense layers of 2000-1000 in each column. (The kernel shape and step size were set to 1, as the sensors were not arranged in a grid.) These were followed by two dense layers of 300 units that combine all the information from the two columns. Layers were initialized from a scaled normal distribution (Glorot & Bengio, 2010), and optimized using the Adam algorithm (Kingma & Ba, 2014) (learning rate $= 0.0001, \beta_1 = 0.999, \beta_2 = 0.999,$ decay $= 0.0001$) on mini-batches of size 20. A dropout rate of 50% was used in the top 3 layers (Srivastava et al., 2014; Baldi & Sadowski, 2014). The *selu* activation (Klambauer et al., 2017) was used in each hidden layer.

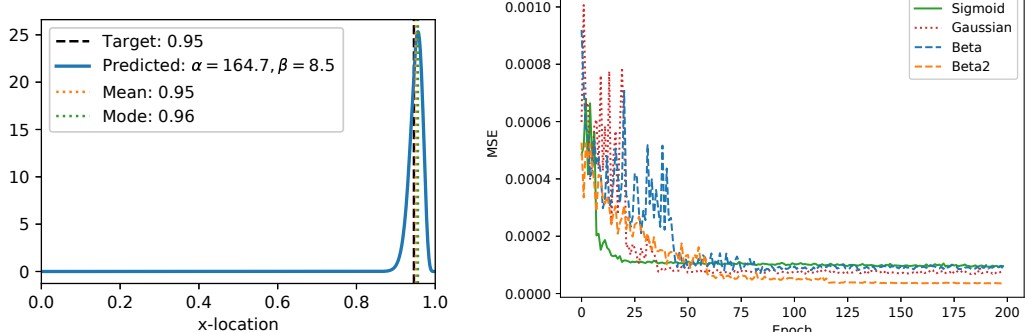

Figure 3: **Left:** Example prediction from the XENON1T network with a Beta output layer. The target is at $0.95$, and the Beta distribution predicted by the network has mean $0.95$ and mode $0.96$. **Right:** Validation set performance on the XENON1T regression task using a linear output layer, a heteroskedastic Gaussian output layer, a Beta output layer, or a Beta output layer with the alternative parameterization (Beta2). Rather than compare the NLL objective, we compare the more intuitive MSE loss, using the means of the Beta distributions as point estimates.

This standard neural network regression model was then compared to three neural networks with more expressive output distributions: (1) a heteroskedastic Gaussian, where the mean was predicted by neurons with linear activation, and the standard deviation was predicted by neurons with exponential activation; (2) a Beta output layer (Equation 11) with exponential activation; (3) and a Beta output layer with the alternative parameterization in Equation 17 where the $\mu$ units have a sigmoid activation and the $\gamma$ units have a exponential activation. The heteroskedastic Gaussian and Beta layers not only provide more informative predictions in the form of distributions that can be used in the downstream analysis, but they also result in *better* performance than the standard approach (Figure 3).

## 4.3 CIFAR-100 TRANSFER TASK

This approach is also appropriate for tasks in which the targets are probabilities. In *model compression*, or *network distillation* (Bucilu et al., 2006; Hinton et al., 2015), a large model (or ensemble of models) is trained for a supervised learning task, and then the information learned is transferred to a separate, smaller model by training the small model to predict the probabilistic output, or "soft targets," of the large model. In many cases, the smaller "student" model will train faster and generalize better than if it had been trained on the actual "hard targets" from the training data set, because there is information, or "dark knowledge", contained in the imperfect predictions of the large model. When the "teacher" model has a sigmoid or *softmax* output, the targets of the student model will be probability values on a multi-dimensional simplex, which can be modelled as samples from a Dirichlet distribution.

We tested this approach by first training a typical convolutional neural network on the CIFAR-100 (Krizhevsky & Hinton, 2009) benchmark data set to serve as a teacher model. The classification data set consisted of 60,000 32-by-32 RGB images from 100 classes, with 50,000 training examples and 10,000 test examples. We used the 18-layer convolutional network architecture from Clevert et al. (2015), with the *selu* transfer function in the hidden layers (Klambauer et al., 2017), a *softmax* output layer, constant dropout rates of (0.0, 0.1, 0.2, 0.3, 0.4, 0.5, 0.0) in each stack, batch size of 100, and the Adam optimizer ($\eta = 0.0001$, $\beta_1 = 0.99$, $\beta_2 = 0.999$). Training was stopped when the validation loss did not improve over 10 epochs. Training samples were augmented using horizontal flipping and random translations of up to 10% vertically or horizontally.

Next, student neural networks were trained to predict the predictions of the teacher network. The student networks had the same architecture and optimization hyperparameters as the original network, except that no dropout regularization was used, and gradients were clipped to a maximum value of 100 for stability. We compared three different types of student network output layers: (1) a standard *softmax* layer with categorical cross-entropy objective, (2) a Dirichlet output consisting of

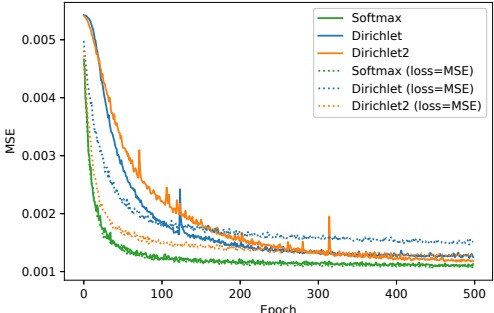

Figure 4: Validation set performance on the CIFAR-100 distillation task using a *softmax* output layer with categorical cross-entropy loss, a Dirichlet output layer as parameterized in Equation 1, or a Dirichlet output layer as parameterized in Equation 7. Rather than compare the NLL objective, we compare the more intuitive MSE loss, using the mean of the Dirichlet distributions as point estimates. Each network was also trained using an MSE objective (using the mean of the Dirichlet as a point-estimate); this had little effect on the *softmax* output, but hurt the final performance of the Dirichlet outputs.

100 $IL$ units as defined in Equation 18, plus a stability factor of $\epsilon = 10^{-6}$, and (3) a Dirichlet with the alternative parameterization given in Equation 7, where the mean of the Dirichlet is specified by a *softmax* of size 100, and the scale is specified with a single $IL$ unit. Figure 4 compares the training trajectories of the networks, and shows that after training, the mean-value point estimate from the Dirichlet output layers have very similar MSE to the predictions of the *softmax* layer. We also include training trajectories for another three networks trained using the MSE objective on the mean-value point prediction from each distribution, rather than the NLL. This initially made training faster for the two versions of the Dirichlet output, but the final performance was not as good; for the *softmax* layer, using the MSE as the objective instead of the cross-entropy made little difference.

### 4.4 LOW-DIMENSIONAL EMBEDDING FOR DIRICHLET-MULTINOMIAL DATA

To test the Dirichlet-multinomial output, we trained an autoencoder network to learn a 2-dimensional embedding of data simulated from high-dimensional, semi-sparse multinomials. This situation is encountered in metagenomics, where the goal is to understand the structure of microbial communities from mixed sequence reads (Holmes et al., 2012).

The data set was constructed by first parameterizing 10 clusters by sampling 10 times from a Dirichlet ($d = 100$, $\boldsymbol{\alpha} = <0.1, 0.1, \ldots, 0.1>$), and then sampling 1,000 times from each cluster, with the number of trials $n$ sampled from the uniform distribution $U(50, 100)$, for a total of 10,000 examples, with each 100-dimensional example consisting of a vector of non-negative counts. Training was performed on 80% of these examples, while 20% was used as a validation set. We trained a neural network autoencoder model consisting of three *tanh* hidden layers of shape 100-2-100, with the 2-dimensional layer serving as the low-dimensional bottleneck, and a Dirichlet-multinomial output layer. The network was trained for 100 epochs, using the Adam optimizer ($\eta = 0.0001$, $\beta_1 = 0.99$, $\beta_2 = 0.999$), batch size of 100, and L2 regularization in the hidden layer (0.0001). As shown in Figure 5, the model has no problem converging to an embedding in which the 10 clusters are clearly separated in the validation set.

## 5 CONCLUSION

In most artificial neural network models, supervised learning corresponds to maximizing the NLL of the training set targets conditioned on the inputs. In this interpretation, each neural network prediction is a *distribution* over possible target values. While the vast majority of neural network classifiers in use today rely on a small set of distributions — the binomial distribution, the categorical distribution, the Gaussian distribution, or the Laplacian distribution — there are many situations for which none of these distributions are appropriate. Here we propose the use of the Beta distribution,

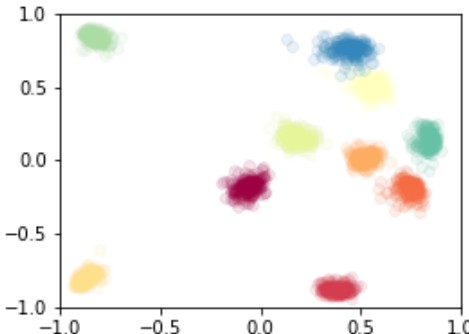

Figure 5: A deep Dirichlet-multinomial autoencoder was used to learn a two-dimensional embedding of simulated samples from 100-dimensional multinomials. The 10 different clusters are readily apparent in the embedding of the validation set examples. The samples shown are colored by their true cluster identity.

Dirichlet distribution, and the Dirichlet-multinomial compound distribution as outputs of neural networks. We show that a neural network can parameterize these distributions and the entire model can be trained using gradient descent on the NLL of the training data targets.

This provides a particularly elegant approach to modelling certain types of network targets. The Beta and Dirichlet provide a better way to model targets that lie on a simplex, such as probabilities or real-values that lie on a bounded interval, and the Dirichlet-multinomial enables us to model vectors of counts using the elegant mathematical properties of the Dirichlet. The predicted distributions have the correct support, so we can use them in decision making and for confidence intervals. Moreover, we have demonstrated through experiments that the expectation over the Dirichlet serves as a good point estimate, with a mean squared error that is similar to optimizing the MSE directly.

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
