# OpenReview forum: "Neural Network Regression with Beta, Dirichlet, and Dirichlet-Multinomial Outputs"
_ICLR.cc/2019/Conference_

### Official Review · AnonReviewer2 · 2018-11-02
**Reasonable proposal and well-written paper, but no new insights and inconclusive empirical results**

**Rating:** 4
**Confidence:** 4

**Review:**

This paper considers parameterizing Dirichlet, Dirichlet-multinomial, and Beta distributions with the outputs of a neural network. They present the distributions and gradients, discuss appropriate activation functions for the output layer, and evaluate this approach on synthetic and real datasets with mixed results. Overall, I found the writing very clear, the main idea sound, and paper generally well executed, but I have serious concerns about the significance of the contributions that lead me to recommend rejection. It would be very useful to me if the authors would provide a concise list of what they consider the main contributions to be and why they are significant. As I see it, the paper does three main things:

1. In section 2, the authors consider parameterizing Dirichlet, Dirichlet-multinomial, and Beta distributions with the outputs of a neural network (Section 2). As the authors note, parameterizing an exponential family distribution with the outputs of a neural network is not a novel contribution (e.g. Rudolph et al. (2016) and David Belanger's PhD thesis (2017)) and though I have never personally seen the Dirichlet, Dirichlet-multinomial, and Beta distributions used, the conceptual leap required is small. Most of section 2 is dedicated to writing down, simplifying, and deriving gradient equations for these three distributions. The simplifications and gradient derivations are well known and appear in many places (e.g. http://jonathan-huang.org/research/dirichlet/dirichlet.pdf, https://arxiv.org/pdf/1405.0099.pdf) and should not be considered contributions in the age of automatic differentiation (see Justin Domke's blog post on autodiff).

2. In section 3, the authors consider the unique challenges of using the proposed networks. They propose targeted activation functions that will improve the stability of learning. I found this to be the most interesting portion of the paper and the most significant contribution. Unfortunately, it is short on details and empirical results are referenced that do not appear in the paper (i.e. the second to last paragraph on page 5). If I were to rewrite this paper, I would focus on answering the question "What are the unique challenges of parameterizing Dirichlet, Dirichlet-multinomial, and Beta distributions with the outputs of a neural network and how can we address them?", replacing section 2 with an expanded section 3.

3. In section 4, the authors evaluate the proposed networks on a collection of synthetic and real tasks. In the end, the results are mixed, with the Dirichlet network performing best on the XENON1T task and the standard softmax network performing best on the CIFAR-100 task. In general, I don't mind mixed results and I appreciate that the authors included both sets of experiments; however, it is important that there is a convincing argument for why one would prefer the proposed solution even when accuracy is the same (e.g. it is faster, it is interpretable, etc.). The authors briefly argue that the proposed methods are superior because they provide uncertainty estimates for the output distributions. This may be true, but they only perform evaluations on tasks where the primary goal is accuracy. If the main benefit of the proposed networks is proper uncertainty quantification, then the evaluations (even if they are qualitative) should reflect that.

In summary, I do not think the models proposed in section 2 are sufficiently novel to justify publication alone which means that the authors need to either: (1) evaluate novel methods that are critical for use of these models or (2) present a convincing evaluation that strongly motivates the proposed model's use or that provides some novel insight into the model's behavior. I think that the authors are on their way to achieving (1), but do not achieve (2). I would suggest finding an application that requires uncertainty estimates for the distribution and centering the paper around that application.

Minor comments:

- Figure 2 (right) should include a y-axis label (e.g. "parameter value").

- In Figure 3 (right), it is not obvious what the "Sigmoid" line corresponds to.

- It is not clear what the authors are trying to show in section 4.1. The EL activation function is smooth and monotone and the likelihood is convex, so there should be no question that the distribution will concentrate around y.

- Section 4.4 was interesting, but would have been more convincing if paired with an evaluation on real data.

---

### Official Review · AnonReviewer3 · 2018-11-03
**No novelty, conceptually problematic, and exceeding the page limit**

**Rating:** 3
**Confidence:** 5

**Review:**

The paper shows how to model the outputs of neural networks via likelihoods other than commonly used ones. The likelihoods discussed include Beta, Dirichlet and Dirichlet-Multinomial. The paper introduces the gradient computation of these likelihoods and test them in several datasets.

This paper lacks novelty and has conceptual mistakes. It is a common practice, in Bayesian learning, to model different types of data with different likelihoods. The examples discussed in this paper are very basis and the gradient computation is standard. I do not see anything new. And the authors misunderstand that if you involve some likelihood in training, you can quantify the uncertainty. It is wrong. Uncertainty should be estimated in the posterior inference framework --- you need to integrate the posterior distribution of the (latent) random variables into the test likelihood to obtain the predictive distribution, from which you can identify the confidence levels. That’s why auto-encoding variational Bayes framework is useful and popular.
What the paper is doing is still the point estimation.

Besides, the paper exceeds the 8-page limit for the content.

---

> ### Author Response · Authors · 2018-11-18
> **Relationship to variational autoencoder models**
>
> Thank you for taking the time to review our paper.
>
> For each input, the proposed model provides a distribution over the possible target values, not just a point estimate. A variational autoencoder is able to model more complex output distributions by replacing a fixed output distribution with a neural network, but it is fundamentally doing the same thing --- it is just another parameterized model trained to maximize the conditional likelihood of the targets. The models described in this paper are simpler and have practical advantages over variational autoencoder models: 1) training can be performed using the true gradients rather than approximations, 2) the form of the output (posterior) distribution is easy to interpret, and 3) it is easy to integrate the output distribution over the target space.

---

### Official Review · AnonReviewer1 · 2018-11-08
**Unoriginal and unfortunately unfocused contributions**

**Rating:** 3
**Confidence:** 4

**Review:**

The authors use neural networks to parameterize conditional probability distributions. This is well-known and has been applied in the literature since extensions to generalized linear models beyond their canonical link function in the 70s. Their transformation from real-valued network output to, say, strictly positive concentration parameters in a Dirichlet are worth studying; but they don't analyze this in any detail.

In addition, while lacking novelty may be fine in and of itself, the purpose of applying these ideas doesn't have a focused purpose. For example, the authors argue in the abstract this quantifies uncertainty. That's only true if you care about data noise, but the end-result is still point estimation for the parameters with uncalibrated probabilities. In the rest of the paper, they write primarily about simplex-valued outputs (i.e., soft one-hot labels).

---

### Public Comment · ~Andrey_Malinin1 · 2018-10-10
**Related work**

Hello!

I find your investigation of the construction and training of models which parameterise the Dirichlet family of distributions to be relevant to our work, especially your investigation into improving the trainability and stability of such models.

In our (due to appear at NIPS 2018 -  https://arxiv.org/pdf/1802.10501.pdf  ) we parameterise a Dirichlet distribution using a DNN in order to derive measures of uncertainty from 'distributions over distributions' for detecting misclassifications and out-of-distribution inputs.

I'm excited by your work and looking forward to any follow up :) .

Best Regards,
Andrey Malinin

---

### Meta-Review · Area_Chair1 · 2018-12-10
**Limited novelty**

**Confidence:** 5
**Recommendation:** Reject

**Metareview:**

This paper proposes to quantify the uncertainty of neural network models with Beta, Dirichlet and Dirichlet-Multinomial likelihood. This paper is clearly written with a sound main idea. However, it is a common practice to model different types of data with different likelihood, although the proposed distributions are not usually used for network output. All the reviewers therefore considered this paper to be of limited novelty. Reviewer 2 also had a concern about the mixed experimental results of the proposed method.

Reviewer 3 raised the concern that this paper did not model the uncertainty of prediction from the uncertainty of the model parameters. It is a common consideration in a Bayesian approach and I encourage the authors to discussed different sources of uncertainty in future revisions.